# Evaluation of Orbital Lymphoproliferative and Inflammatory Disorders by Gene Expression Analysis

**DOI:** 10.3390/ijms23158609

**Published:** 2022-08-03

**Authors:** Karim Al-Ghazzawi, Sven Holger Baum, Roman Pförtner, Svenja Philipp, Nikolaos Bechrakis, Gina Görtz, Anja Eckstein, Fabian D. Mairinger, Michael Oeverhaus

**Affiliations:** 1Department of Ophthalmology, University Hospital Essen, 45147 Essen, Germany; karim.alghazzawi@icloud.com (K.A.-G.); svenja.philipp@uk-essen.de (S.P.); nikolaos.bechrakis@uk-essen.de (N.B.); gina.goertz@uk-essen.de (G.G.); anja.eckstein@uk-essen.de (A.E.); 2Department of Oral and Maxillofacial Surgery, University of Duisburg-Essen, Kliniken-Essen-Mitte, 45136 Essen, Germany; dr-baum@gmx.de (S.H.B.); r.pfoertner@kem-med.com (R.P.); 3Institute of Pathology, University Hospital Essen, University of Duisburg-Essen, 45147 Essen, Germany

**Keywords:** idiopathic orbital inflammation, MALT, lymphoma, IgG4-ROD, non-specific orbital inflammation, pseudotumor orbitae, IgG4-related orbitopathy

## Abstract

Non-specific orbital inflammation (NSOI) and IgG4-related orbital disease (IgG4-ROD) are often challenging to differentiate. Furthermore, it is still uncertain how chronic inflammation, such as IgG4-ROD, can lead to mucosa-associated lymphoid tissue (MALT) lymphoma. Therefore, we aimed to evaluate the diagnostic value of gene expression analysis to differentiate orbital autoimmune diseases and elucidate genetic overlaps. First, we established a database of NSOI, relapsing NSOI, IgG4-ROD and MALT lymphoma patients of our orbital center (2000–2019). In a consensus process, three typical patients of the above mentioned three groups (mean age 56.4 ± 17 years) at similar locations were selected. Afterwards, RNA was isolated using the RNeasy FFPE kit (Qiagen) from archived paraffin-embedded tissues. The RNA of these 12 patients were then subjected to gene expression analysis (NanoString nCounter^®^), including a total of 1364 target genes. The most significantly upregulated and downregulated genes were used for a machine learning algorithm to distinguish entities. This was possible with a high probability (*p* < 0.0001). Interestingly, gene expression patterns showed a characteristic overlap of lymphoma with IgG4-ROD and NSOI. In contrast, IgG4-ROD shared only altered expression of one gene regarding NSOI. To validate our potential biomarker genes, we isolated the RNA of a further 48 patients (24 NSOI, 11 IgG4-ROD, 13 lymphoma patients). Then, gene expression pattern analysis of the 35 identified target genes was performed using a custom-designed CodeSet to assess the prediction accuracy of the multi-parameter scoring algorithms. They showed high accuracy and good performance (AUC ROC: IgG4-ROD 0.81, MALT 0.82, NSOI 0.67). To conclude, genetic expression analysis has the potential for faster and more secure differentiation between NSOI and IgG4-ROD. MALT-lymphoma and IgG4-ROD showed more genetic similarities, which points towards progression to lymphoma.

## 1. Introduction

Different non-infectious inflammatory orbital diseases have been described as ‘orbital pseudotumor’ in the past, due to their similar clinical presentation. By means of the evolving histopathological and molecular analysis of biopsies, more and more entities were excluded from this broad diagnosis (e.g., vasculitis, lymphoproliferative disorders) [1]. Despite still being commonly used, an orbital pseudotumor has been widely replaced by labels reflecting better the clinical and histopathological aspects of the disease: non-specific orbital inflammation (NSOI), idiopathic orbital inflammation (IOI) and ‘idiopathic non-granulomatous orbital inflammation’ have been suggested, among others, instead of the old terminology [2,3,4]. In the following, the terminus “NSOI” will be used for the sake of simplicity, defining it as an inflammatory process of the orbit without underlying systemic or local cause. It is still a diagnosis with heterogeneous clinical and histological presentations and remains a diagnosis of exclusion [4]. Typically, patients present with unilateral exophthalmos, pain, periorbital edema and erythema, as well as diplopia [5]. Since these symptoms are unspecific, they cannot be used for robust diagnosis [6]. Due to the heterogeneous nature of the disease, NSOI can be subcategorized with regard to onset of symptoms, location and histopathology [1,3,4,5]. Before diagnosing NSOI, the clinician has to mind numerous clinical similar diagnoses (e.g., IgG4-related orbital disease, orbital lymphoma, sarcoidosis); therefore, most patients need an orbital biopsy even after radiological imaging [7]. Despite all the advances in histopathological techniques (e.g., immunostaining), the differentiation between NSOI and IgG4-related orbital disease (IgG4-ROD) remains challenging [1,8,9]. Furthermore, some patients with NSOI respond very well to steroids, whereas others need multiple steroid sparing therapies to prevent relapses and maintain a stable state. Therefore, we aimed to distinguish between these different autoimmune orbital diseases, and NSOI in particular, depending on their clinical course, at the molecular level. Methodically, we used NanoString nCounter technology, which allows high-throughput, precise and reliable RNA analysis, even of formalin-fixed, paraffin-embedded tissue (FFPE) [10]. Previous studies have shown that FFPE tissue can be used as a feasible source for gene expression analysis and delivers similar results compared to fresh-frozen tissue [11]. This allowed us to use routinely acquired and stored orbital biopsies, which is a great advantage for elucidation of such a rare disease. The technology detects abnormally altered genes or molecular pathways and is therefore an ideal tool for the gene expression and transcriptome analysis of these orbital diseases to enable better diagnostic differentiation in the future. Furthermore, we aimed to elucidate the mechanisms for progression of these chronic inflammatory diseases to the most common orbital neoplasm: MALT (mucosa-associated tissue) lymphoma. Therefore, we analyzed all orbital inflammatory diseases retrospectively and selected three patients for each group for a broad RNA expression analysis and detection of marker genes. Secondly, we analyzed a larger cohort regarding the initially found genes to validate the results. 

## 2. Results

### 2.1. Study Population

Among the 60 patients, the mean age was 59.4 ± 16.7 years and 24 (42%) were female. We initially analyzed 12 acute inflamed biopsy specimens from 12 patients, including 3 suffering from IgG4-ROD, 3 with uncomplicated NSOI, 3 with relapsing NSOI and 3 orbital MALT-lymphoma patients. The validation cohort comprised of 48 independent patients: 11 IgG4-ROD, 13 MALT lymphoma and 24 NSOI (Table 1). Statistical analysis for age between groups showed that the NSOI patients were significantly younger compared to the lymphoma patients (ANOVA with Holm–Šídák’s multiple comparisons test, adjusted *p* = 0.0012), but not compared to IgG4-ROD patients (*p* = 0.07). IgG4-ROD and MALT also showed no significant age difference (*p* = 0.27). This is in accordance with previous publications [4,7].

### 2.2. Clinical Examination

Eyelid swelling, pain, redness, proptosis, ptosis, limited ocular motility, corneal dryness and visual dysfunction were the typical clinical presentations of all patients, showing variety according to the amount and type of orbital tissue involved. The patients showed across entities similar clinical signs without significant differences (Figure 1). 

### 2.3. Expression Analysis

Gene expression analysis was successful in all 12 samples. After biological and technical normalization, 370 (29.3%) out of 1263 genes were identified as transcripts with relevant gene expression in at least one of the three groups, NSOI, lymphoma and IgG4-ROD, respectively. For each subgroup, significantly altered genes were extracted and investigated. 

Comparing NSOI with all other groups, 13/370 (3.8%) genes showed significant deregulation after adjustment. Of those, 12 showed upregulation in NSOI (see Table 2) and one presented with downregulation (*RPS27A*) (see Appendix A for detailed information). For the lymphoma group, 247/370 (66.8%) showed significantly altered gene expression levels compared to all others (see Table 2 for the top 20 differentially expressed genes) Noticeable, all those genes showed reduced expression levels (Appendix A for details on all genes). Furthermore, for the IgG4-ROD group, 62/340 (18.2%) relevantly expressed genes show altered gene expression levels compared to the other groups, including 61 upregulated and one downregulated target expression (see Table 2 for the top 20 differentially expressed genes and Appendix A for all). Interestingly, there was a relatively large overlap between genes differentially expressed in NSOI and lymphoma (10/13 (76.9%) NSOI genes and 10/247 (4%) of lymphoma genes) or IgG4-ROD and lymphoma (39/62 (62.9%) IgG4-ROD genes and (39/247 (15.4%) of lymphoma genes), whereas NSOI and IgG4-ROD showed only 1 overlap (7.7% of NSOI and 1.6% of IgG4-ROD genes, Figure 2). Those overlaps are mainly based on the significant downregulation of genes in the lymphoma group and specific upregulation in the two other respective groups. 

### 2.4. Decision-Tree-Based Analysis of Differences between Groups 

Decision-tree-based analysis using the “conditional inference tree” (CIT) machine learning algorithm regarding the different groups revealed a three-tier system based on the *PLA2G2A*, *RBM47* and *AQP1* expression levels. Looking at the *PLA2G2A* expression and its best cut-off (calculated cut-off: 610 counts, *p* = 0.017), the group with the *PLA2G2A* overexpression consisted only of severe NSOI. Furthermore, in those samples where *PLA2G2A* expression was beneath the cut-off, the expression of *RBM47* (calculated cut-off: 38 counts, *p* = 0.024) identified the subgroup of IgG4-ROD. On the other hand, in samples below *PLA2G2A* and *RBM47*, the expression cut-off, the expression of *AQP1* (calculated cut-off: 30 counts, *p* = 0.026) divided them into the two separate subgroups comprising lymphomas and mild NSOI, respectively (Figure 3). 

Furthermore, comparisons for each single group were calculated. For NSOI, CIT revealed a one-tier system based on *AQP1* expression, with the appertaining cut-off of >138 counts identifying NSOI cases (*p* = 0.011; Appendix A). 

For lymphoma, CIT again revealed a one-tier system based on *RAC1* expression, with the appertaining cut-off of ≤193 counts identifying lymphoma cases (*p* = 0.007; Appendix A). For IgG4-ROD, CIT also revealed a one-tier system based on *IRF4* expression, with the appertaining cut-off of >473 counts identifying IgG4-ROD cases (*p* = 0.003; Appendix A).

### 2.5. Calculation of Scores

Based on these findings, a scoring system for each subgroup was created. The score identifying NSOI consists of 12 genes, where upregulation was scored for 11 (*ANG*, cut-off: 60; *AQP1*, cut-off: 125; *BMPR1A*, cut-off: 125; *CALD1*, cut-off: 300; *ERBB2*, cut-off: 50; *ITGA7*, cut-off: 50; *NME4*, cut-off: 50; *NR4A3*, cut-off: 50; *PDCL3*, cut-off: 100; *TNXB*, cut-off: 75; *SDC4*, cut-off: 100) and downregulation for 1 of those (*RPS27A*, cut-off: 5500). Generalized linear modelling revealed a *p* < 0.001 for this scoring system. Subsequently, the relative risk for this respective subtype was calculated for all possible score numbers. Based in these data, logistic regression in the form y ~ 1/(1 + exp(x* − a + b)) was used for generating the model. This resulted in an estimate for a of 15.4536 (*p* < 0.0001) and b of 131.3080 (*p* < 0.0001)

The score identifying lymphoma also consists of 12 genes, where downregulation was scored for all of them (*APP*, cut-off: 400; *AQP1*, cut-off: 50; *CD59*, cut-off: 500; *CLIC4*, cut-off: 200; *HSPB1*, cut-off: 300; *JAG1*, cut-off: 50; *LGALS3*, cut-off: 200; *MEG3*, cut-off: 75; *NRP1*, cut-off: 250; *RAC1*, cut-off: 250; *RTN4*, cut-off: 250; *TNS1*, cut-off: 150). Generalized linear modelling revealed a *p* < 0.001 for this scoring system.

Subsequently, the relative risk for this respective subtype was calculated for all possible score numbers. Based on these data, logistic regression in the form y ~ 1/(1 + exp(x* − a)) was used for generating the model. This results in an estimate for a of 8.5000 (*p* = 0.0072).

The score identifying IgG4-ROD consists of 15 genes, where upregulation was scored for all of them (*ADAM9*, cut-off: 250; *ADAM17*, cut-off: 300; *ANXA2P2*, cut-off: 3000; *CLEC2B*, cut-off: 500; *CLIC4*, cut-off: 500; *FCGR2A*, cut-off: 500; *GPX1*, cut-off: 600; *NRP1*, cut-off: 1200; *PSMD7*, cut-off: 550; *RBX1*, cut-off: 500; *RHOA*, cut-off: 700; *RNH1*, cut-off: 350; *TCEB1*, cut-off: 450; *TLR4*, cut-off: 450; *TNFRSF1A*, cut-off: 400). Generalized linear modelling revealed a *p* < 0.001 for this scoring system.

Subsequently, the relative risk for this respective subtype was calculated for all possible score numbers. Based on these data, logistic regression in the form y ~ 1/(a + exp(x* − b)) was used for generating the model. This results in an estimate for a of 0.8164 (*p* = 0.0099) and b of 11.831 (*p* = 0.0003). Scores for each entity were validated using leave-one-out cross validation. Graphs of all the different scores regarding subtype are depicted in Figure 4.

### 2.6. Data Validation

To check the biological and clinical reliability and meaningfulness of the calculated multi-parameter scoring systems, a validation cohort of 48 independent samples was analyzed for determination of prediction accuracy. As expected, performance within the validation cohort was reduced compared to the training data set, but, especially for IgG4 and MALT lymphoma, still reaches statistical significance and shows good performance with high accuracy. For the IgG4 score, analysis of variance (ANOVA) of the binomial generalized linear regression (logit) resulted in a *p* = 0.0051, with an AUC in the ROC performance plot of 0.81 (Figure 5A). The lymphoma score performed slightly better than the IgG4 score, showing an AUC of 0.82 with *p* < 0.0001.

In synopsis of all three scores (illustrated in Figure 5C), the nice and hard discrimination between IgG4 and MALT lymphoma was validated. For NSOI, the used models performed worse but still reached statistical significance, showing an association of higher scores with a higher probability of NSOI (*p* = 0.0167, AUC: 0.67; Appendix A). Four NSOI cases showed a lymphoma score of 1 or higher. Reevaluation of those four cases revealed disease progression and reclassification as MALT lymphomas, whereas the remaining 20 NSOIs did not show this (until now). 

## 3. Discussion

The present study provides the first differential RNA analysis for specific target genes of orbital inflammatory diseases. We showed that IgG4-ROD and NSOI can also be distinguished by RNA expression analysis using a multiparameter scoring algorithm. Furthermore, we could elucidate the genetic similarities between these orbital inflammatory diseases and MALT lymphoma, which might lead to a better understanding of the progression of these diseases to lymphoma. In the future, genetic profiling of biotic specimens could lead to a more secure and faster diagnosis of orbital lesions and thus to a faster and more effective treatment. 

### 3.1. Clinical Examination

Our study population showed, as described before, varying clinical manifestations for NSOI, IgG4-ROD and orbital lymphoma. The age and gender distributions are in accordance with previous studies, showing a significantly higher age for lymphoma patients compared to the inflammatory orbital disorders, with the lowest age for NSOI [4,7,12,13]. Pain remains a major symptom and a diagnostic criterion for NSOIs [6]. As described before, the pain varied from discomfort and tenderness to severe pain on eye movement in the periorbital region. Severe NSOIs showed significance in orbital pain as a symptom when compared to the other entities (Figure 1). This in accordance with previous publications, which showed less pain and less impaired ocular motility in patients with IgG4-ROD [14]. Motility showed in our small study cohort no significant differences. Proptosis, diplopia and eyelid swelling were present in all entities to about the same extent. The patient cohort showed, as in previous studies, an overlap of clinical findings in these orbital diseases. Therefore, imaging and mostly biopsies are still necessary for a safe diagnosis. 

### 3.2. Gene expression Analysis as Diagnostic Tool

Recent reports revealed the difficulty in distinguishing NSOIs from IgG4-ROD by routinely used forms of diagnostics, such as histopathology and immunohistochemistry: IgG4-ROD is characterized by dense lymphoplasmacytic infiltrate, storiform fibrosis and obliterative phlebitis. NSOIs tend to manifest in a variable histopathological picture ranging from a lack of the previously mentioned characteristics to highly inflamed infiltrates and storiform fibrosis, depending on time of biopsy, previous therapy and diagnosis [1,15]. As lymphocytes are short lived (a life span of several weeks to month), a diagnostic trial with corticosteroids on these entities before diagnostic surgery may mask histopathological presentation. Approximately one third of IgG4-ROD do not meet the criteria and are therefore diagnosed as possible IgG4-ROD. This often results in confusion and delayed diagnosis [15,16,17,18]. Therefore, diagnostic criteria for both NSOI and IgG4 based on histopathology remain critical and less accurate when used at different time points due to practical considerations regarding specimen acquisition and diagnostic trials on corticosteroids. Therefore, we aimed to amend the diagnostic possibilities by evaluating the RNA expression. Gene expression profiling can already help distinguish different causes of synovitis, esophagitis, myocarditis and uveitis [19,20,21,22]. Our main goals proceeding with this study were the identification of differential gene expression in at least one of the groups and the evaluation of expression clusters and establishment of a set of marker genes as a potential diagnostic and prognostic tool. This is to date the first comparative RNA expression analysis of these entities, although Higgs et al. (2017) analyzed serum of IgG4-RD serum but not of IgG4-ROD tissue [23]. Taylor et al. (2019) already showed that genome sequencing (NGS) can help to distinguish otherwise difficult differentiation between IgG4-RD and inflammatory fibroblastic tumors and is a useful tool for molecular diagnosis [24]. Asaskage et al. (2020) demonstrated by high-throughput RNA sequencing of biopsy specimen that IgG4-ROD shows 35 upregulated genes compared to healthy controls and reactive lymphoid hyperplasia. These included matrix metallopeptidase 12 (*MMP12*) and secreted phospoprotein 1 (*SPP1*), which were proposed as new biomarkers [25]. Our analysis showed upregulation of MMP14 in IgG4-ROD, which highlight this pathway, and which should be further explored. In our cohort, the gene expression analysis revealed also significant up- and downregulated genes in the IgG4-ROD biopsy specimen, which could be used to reliably distinguish them from NSOI and MALT lymphoma. 

Rosenbaum et al. (2017) analyzed the gene expression of inflamed lacrimal glands in different diseases, including NSOI, and could demonstrate that biopsy specimens showed a clear heterogeneity between entities [26]. However, 32% of NSOI specimens could not be distinguished from healthy controls, which might be due to previous prednisone treatment, which is known to have a large effect on gene expression [27]. In our cohort, the differences in significantly upregulated and downregulated genes being integrated into the “conditional inference tree” (CIT) machine learning algorithm helped us generate a specific probability calculation tool for each entity based on potential target genes. By employing this decision tree-based analysis, we could identify a set of marker genes for each entity out of the 1364 analyzed genes. In the following, we used the 35 identified biomarker genes to validate the scoring algorithm in a larger sample size. Here, we could show that especially for IgG4-ROD and MALT lymphoma the scoring algorithm is highly reliable and effective. For NSOI, the score worked less well but was still significant. This reflects the heterogeneous nature of the NSOI patients and points toward the suspected sub-entities of NSOI. Therefore, we plan further RNA expression analysis of NSOI to further distinguish possible molecular sub-entities and improve the score further. Still, our multiparameter scoring algorithm based on the identified marker genes can already be used for a fast and more accurate diagnosis of IgG4-ROD, NSOI and MALT lymphomas. Thus, our marker genes might reduce the morbidity of patients since an adequate therapy could be faster applied, thereby reducing the possible long-term damages of these chronic diseases. An integration of molecular marker-based approaches into clinical decision making could significantly improve clinical management of individual patients, hopefully resulting in a generally improved strategy for those cases that are nowadays clinically difficult to resolve. 

Our analysis showed distinct genetic differences between NSOI and IgG4-ROD and lymphoma patients. However, the distinction of relapsing NSOI and milder cases was not possible with gene expression analysis. Due to the heterogeneous nature of the disease, which was also shown by Rosenbaum et al., we must assume that there is no simple gene expression variance that explains the course of the disease. Larger expression profiling analyses might further elucidate a possible connection. Furthermore, single-cell transcriptomics might even lead to a further understanding of the underlying pathophysiology, as in rheumatoid arthritis [28]. In theory, genetic expression analysis of specific subtypes of the three entities could be correlated with the clinical findings and therapeutic effect to enable more effective treatment strategies. Until now, no correlation between recurrence rate and histological subtype has been found [4,29].

### 3.3. Progression to MALT Lymphoma

Until now, the progression of chronic orbital inflammatory diseases into lymphoid, monocular proliferation remains unclear. Reports suggest that 12% of orbital MALT lymphoma derive from IgG4-ROD [30,31]. This could explain why the differentiation between IgG4-ROD and MALT is sometimes difficult, since both share similar immunophenotypic profiles [17] and histological features [18], such as IgG4-positive cells (though less in MALT) and T cell infiltration (higher in IgG4 as in MALT) [32].

Our RNA expression analysis showed marked differences between the three entities, as well as similarities: differential gene expression analysis revealed a much larger overlap between lymphoma and IgG4-ROD as well as NSOI, whereas IgG4-ROD and NSOI shared only one gene. Previous publications suggested an overlap or modest link between IgG4-ROD and NSOI, which could not be verified in our study cohort [16,33,34]. In contrast, the suggested progression of chronic inflammation, as in IgG4-ROD and NSOI, to orbital MALT lymphoma seems reasonable since both entities showed a significant overlap in differentially expressed genes (Figure 2). 

Shimizu et al. (2021) introduced a differentiation by metabolic signatures for IgG4-ROD vs. MALT lymphoma [35]. Their cluster analysis did not only propose a clear differentiation but also showed similarities for both entities in some patients. They also propose a “class–switch” of metabolism through the process [30,31]. Both analyses undermine the hypothesis of “chronic inflammation deriving to two different entities”. For MALT lymphoma, this “switch” could be a continues proliferation of malignant B cells, whereas for IgG4-ROD it is an infiltration of lymphocytes and IgG4-producing plasma cells. This confirms the need for biological markers in early stages of the disease. Our results could be used for further investigations of this connection and might even lead to a risk score to estimate the progression risk of chronic orbital inflammation. 

### 3.4. Limitations

Limitations of this study include its relatively small number of patients that were collected from only one institute and its retrospective design. This might have resulted in selection and confounding bias. Furthermore, the heterogeneous histopathological findings and orbital locations might mask gene expression differences that are only present in specific subtypes of NSOI. However, since the main goal of this study was to differentiate between NSOI and IgG4, this broader analysis revealed the most prominent genes for each entity. Future studies might further elucidate the gene expression patterns of histopathological subtypes and different locations within the orbit. This might even lead to finding a new entity within NSOIs, as has happened before with IgG4-ROD. 

## 4. Materials and Methods

### 4.1. Study Population

In this study, we retrospectively analyzed all patient records with an acute orbital inflammatory mass who visited our tertiary referral orbital center between 2000 and 2020 and were diagnosed after orbital biopsy as either NSOI, IgG4-ROD or orbital MALT lymphoma. Only patients with complete data sets were considered for further molecular analysis. First, we identified patients with a typical clinical course and certain diagnosis for NSOI, relapsing NSOI, IgG4-ROD and MALT lymphoma patients (*n* = 3 each) for a broad RNA expression analysis and screening for potential biomarker genes. For further validation, a larger cohort comprised of 48 independent patients was analysed: 11 IgG4-ROD, 13 MALT lymphoma and 24 NSOI samples.

### 4.2. Clinical Examination

All patients showed a new onset of inflammation, varying accordingly to extent in orbit and location, ranging from space-occupying or infiltrating lesions with proptosis, motility dysfunction to severe orbital inflammation with pain edema and redness. First, all patients were evaluated by a highly trained orthoptist and afterwards by a specialized ophthalmologist (A.E., M.O.). Eye examinations included slit-lamp biomicroscopy, applanation tonometry, fundoscopy, Hertel and Naugel exophthalmometry, assessment of subjective diplopia and objective measurement of misalignment using the prism-cover test, and measurement of monocular excursions and visual acuity. The clinical examination included evaluation of eyelids (ptosis, retraction, swelling, erythema), orbit (proptosis, palpable mass), globe (injection, chemosis, intraocular inflammation, retinal abnormality), and optic nerve function (relative afferent pupillary defect, color vision, visual field, visual acuity). These examinations aimed to characterize the inflammation, anatomic location and functional implications. By this, further diagnostic procedures were determined (e.g., serologic parameter) to rule out other differential diagnoses such as Graves’ orbitopathy (GO) and vasculitis (granulomatosis with polyangiitis, GPA). In synopsis with MRI images, a tentative diagnosis was determined before orbital biopsy was performed. The diagnoses of NSOI, IgG4-ROD and orbital MALT lymphoma were finally based on clinical, flow cytometric and histological (including immunostaining) examinations. IgG4-ROD was diagnosed in accordance with the published criteria [16]. Briefly, IgG4-ROD was diagnosed in the presence of (1) enlargement of orbital tissues with marked lymphoplasmatic infiltration and fibrosis/sclerosis; and (2) >50 IgG4 positive plasma cells per high power field (IgG4+/IgG Ratio > 40%) and a serum IgG4 level > 135 mg/dL. 

Patients who showed persistent or relapsing inflammation after two prednisolone courses were defined as relapsing NSOI. They underwent as a follow-up anti-inflammatory therapy another high-dose and prolonged prednisolone or other immunosuppressive therapy (e.g., methotrexate, azathioprine) to prevent therapeutic recurrences of NSOIs.

### 4.3. Tissue Samples

We obtained 60 extraconal lesion biopsy specimens from 60 individuals: 27 non-specific orbital inflammation (NSOI), 14 IgG4-related ophthalmic diseases (IgG4-ROD) and 16 MALT lymphoma patients. All tissue samples were routinely processed, formalin fixed, and paraffin embedded. 

### 4.4. RNA Extraction

One to three paraffin sections with a thickness of 7 μm per sample was deparaffinized with xylene prior to RNA extraction using the RNeasy FFPE kit (Qiagen, Hilden, Germany) according to the manufacturer’s recommendations with slight adjustments. Total RNA concentrations were measured using a Nanodrop 1000 instrument (Thermo Fisher Scientific, Waltham, MA, USA) [36].

### 4.5. nCounter CodeSet Design and Expression Analysis

Multiple genes involved in tumor- and inflammation-associated pathways were selected based on the current literature in order to screen for biological references potentially/probably distinguishing NSOIs from the other lesions and additionally give pathophysiological insight into each entity.

Gene expression patterns were screened for prognostic and predictive biomarkers using the NanoString nCounter platform for digital gene expression analysis with the appurtenant PanCancer Progression Profiling panel, consisting of 770 tumor-related genes and the Immunology V2 Profiling panel consisting of 594 genes mediating immune response as well as 30 reference genes (see Appendix A), experiment reagents were designed and synthesized by NanoString Technologies (Seattle, WA, USA). Hybridizations were performed using the high-sensitivity protocol on the nCounter Prep-Station. Post-hybridization processing was performed by using the nCounter MAX/FLEX System (NanoString) and the cartridge was scanned using a Digital Analyzer (NanoString). The cartridge was read with maximum sensitivity (555 FOV). A 100-ng sample input was used for each reaction.

### 4.6. Nanostring Data Processing

NanoString data processing was done with the “R i386 statistical programming environment” (v4.0.3), R Foundation for Statistical Computing, Institute for Statistics and Mathematics, Vienna, Austria. Considering the counts obtained for positive control probe sets, raw NanoString counts for each gene were subjected to a technical factorial normalization, carried out by subtracting the mean counts plus two times the standard deviation from the CodeSet inherent negative controls. Subsequently, a biological normalization using the included RNA reference genes was performed. Additionally, all counts with *p* > 0.05 after one-sided *t*-test versus negative controls plus 2× the standard deviation were interpreted as not expressed to overcome basal noise [37].

### 4.7. Statistical Evaluation

Statistical analysis was carried out using the “R i386 statistical programming environment” (v4.0.3). Prior to exploratory data analysis, the Shapiro–Wilks test was applied to test for a normal distribution of each data set for ordinal and metric variables. Resulting dichotomous variables underwent either the Wilcoxon Mann–Whitney rank sum test (non-parametric) or a two-sided student’s *t*-test (parametric). For comparison of ordinal variables and factors with more than two groups, either the Kruskal–Wallis test (non-parametric) or ANOVA (parametric) were used to detect group differences. Double dichotomous contingency tables were analyzed using Fisher’s exact test. To test the dependency of the ranked parameters with more than two groups, the Pearson’s Chi-squared test was used. Correlations between metrics were tested applying Spearman’s rank correlation test as well as Pearson’s product–moment correlation testing for linearity. Quality control of the run data was first performed basically by mean-vs-variances plotting to find the outliers at the target or sample level. True differences and clusters at both the target and sample level were calculated by correlation matrices analysis. To further specify the different candidate patterns, both unsupervised and supervised clustering as well as principal component analysis were performed to overcome commonalities and differences. Sensitivity and specificity of markers were determined from receiver operating characteristic (ROC) curves illustrating their performance to discriminate the studied groups. The bootstrap procedure (1000 iterations) was used for internal validation of the estimates in the ROC analyses. The best candidate genes were selected and binarized (0, 1; with 1 equaling a better chance of an event) by their respective cut-offs and finally summed up. Robustness of the generated scores was validated using generalized linear modeling. The resulting scores were compared with respect to sensitivity and specificity. The probability for each entity was determined using the nonlinear (weighted) least-squares estimates of the parameters of a nonlinear fitted regression model [38,39]. Adaption of profiles for diagnostic purposes were modeled with the supervised machine learning tool conditional interference trees (CTree), as implemented in the “party” library of R [40] using leave-one-out cross-validation. CTree is a non-parametric class of regression tree leading to a non-parametric class of tree-structured regression models, embedding a conditional inference procedure, applicable to all kinds of regression problems, including nominal, ordinal, numeric, censored as well as multivariate response variables and arbitrary measurement scales of the covariate [40]. Due to the multiple statistical testing, the *p*-values were adjusted by using the false discovery rate (FDR). The level of statistical significance was defined as *p* ≤ 0.05 after adjustment.

### 4.8. Model Validation

For validation of our results obtained in the screening collective, an independent validation cohort (*n* = 48) comprising 11 IgG4-ROD, 13 MALT lymphoma and 24 NSOI samples were examined. Gene expression pattern analysis of the target genes encompassing all three models was performed using a custom-designed CodeSet comprising 35 target genes as well as five reference genes (*ACTB*, *B2M*, *GAPDH*, *RPL19*, *RPLP0*) previously identified as being stably expressed in the screening cohort. All CodeSets along with the experiment reagents were designed and synthesized by NanoString Technologies (Seattle, WA, USA). In line with the methodology described above, post-hybridization processing was performed using the nCounter MAX/FLEX System (NanoString) and cartridges were scanned on a Digital Analyzer (NanoString). Samples were analyzed on the NanoString nCounter PrepStation, using the high-sensitivity program, and cartridges were read at maximum sensitivity (555 FOV). Again, a 100-ng total RNA input was used for each reaction, raw data processing as well as its analysis have been performed in the same way as described for the initial screening.

## 5. Conclusions

Through our study we could elaborate that, although NSOIs, IgG4-ROD and orbital MALT lymphoma share genes as well as clinical symptoms, they can be safely distinguished due to their distinctive genetic expression patterns by implementing significantly upregulated and downregulated genes in a machine learning algorithm. The identified genes could be used as biomarkers to simplify the differential diagnosis of these lesions, since histopathological and immunohistochemical findings can be inconclusive.

## Figures and Tables

**Figure 1 ijms-23-08609-f001:**
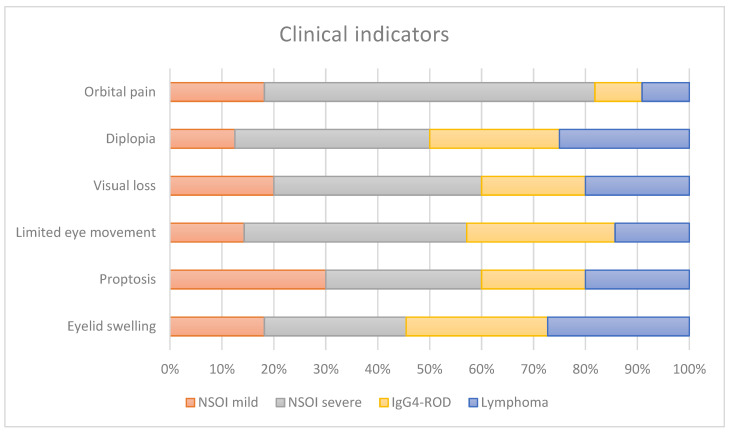
Clinical symptoms present in our index population stratified for each disease entity and subtype of disease.

**Figure 2 ijms-23-08609-f002:**
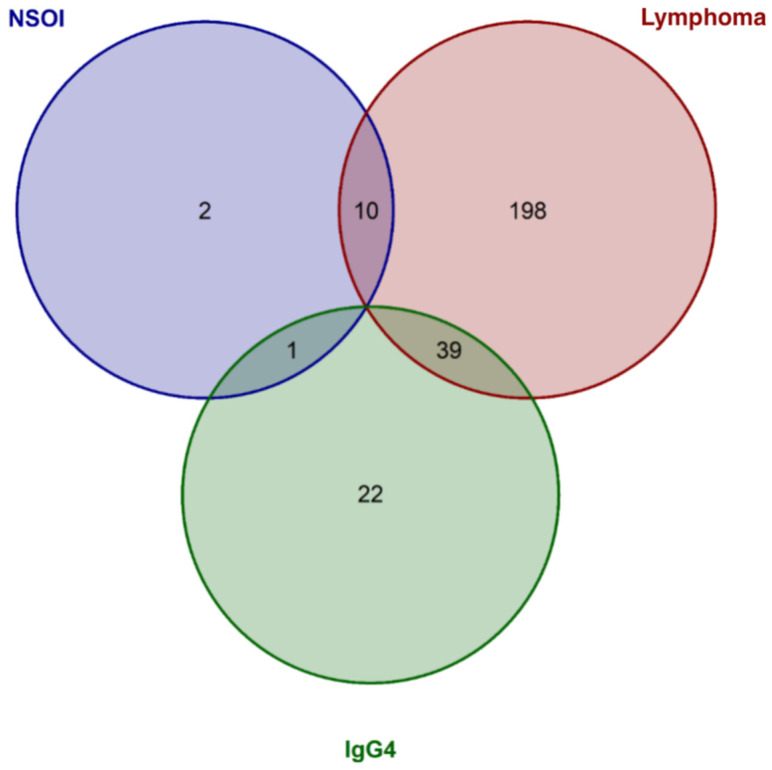
Venn diagram showing the significantly expressed genes for each group and the overlap between the entities. Whereas NSOI and lymphoma as well as IgG4 and lymphoma showed many overlaps, only one overlap could be identified for IgG4 and NSOI.

**Figure 3 ijms-23-08609-f003:**
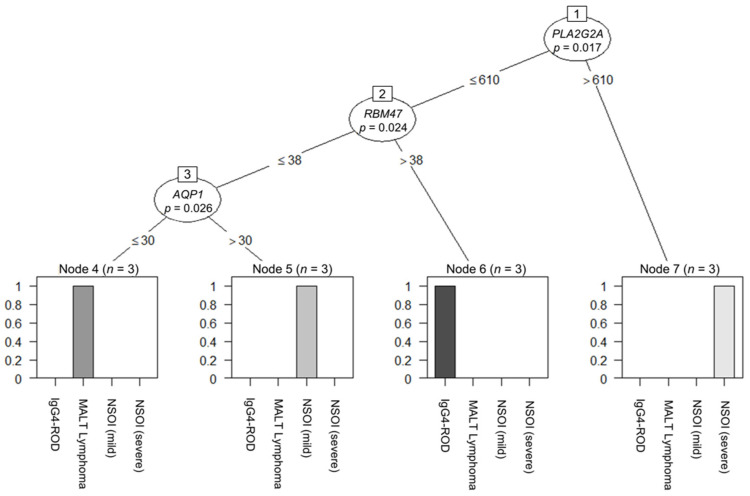
Decision-tree-based analysis using the “conditional inference tree” (CIT) machine learning algorithm regarding the different groups revealed a three-tier system based on (1) *PLA2G2A* (*p =* 0.017), (2) *RBM47* (*p =* 0.024), and (3) *AQP1* (*p =* 0.026) expression levels.

**Figure 4 ijms-23-08609-f004:**
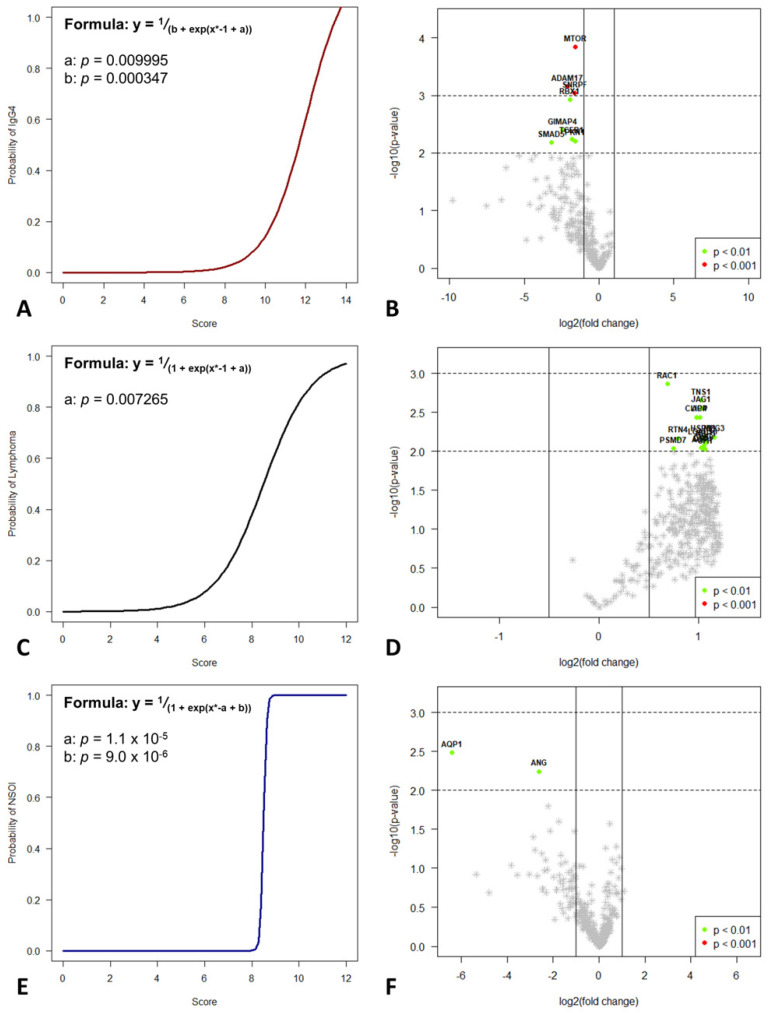
Decision-tree-based analysis using the “conditional inference tree” (CIT) machine learning algorithm revealed significant differences between the three groups. Based on these differently up- and downregulated genes, a logistic regression model was used to differentiate IgG4-ROD (**A**), MALT lymphoma (**C**), and NSOI (**D**) with a high probability. Volcano plots for IgG4-ROD (**B**), MALT lymphoma (**E**), and NSOI (**F**) shows the most significantly differentially expressed genes.

**Figure 5 ijms-23-08609-f005:**
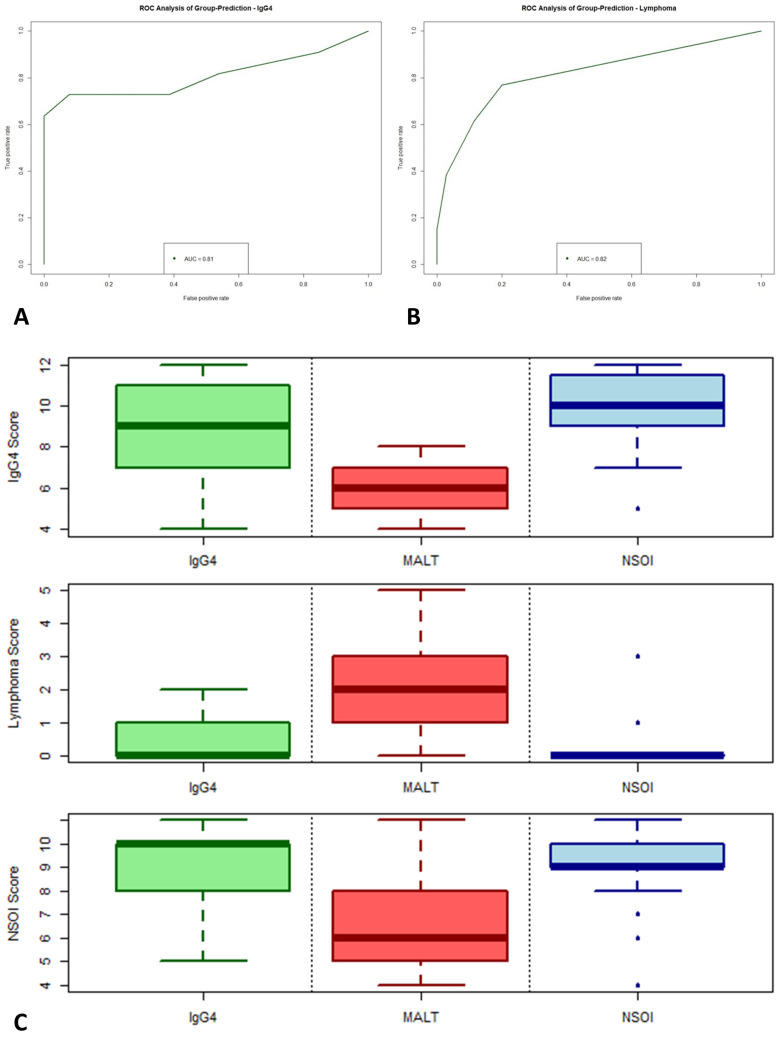
Validation of our logistic regression model based on the differently up- and downregulated genes. The validation cohort compromises 48 independent samples. ROC analysis of group-prediction for IgG4-ROD (**A**) and MALT lymphoma (**B**). Synopsis of all three scores (**C**).

**Table 1 ijms-23-08609-t001:** Demographic features of patients with non-specific orbital inflammation, IgG4-related orbital disease and orbital MALT lymphoma in our RNA expression analysis study.

	NSOI	IgG4-ROD	MALT Lymphoma	*p*
Number	30	14	16	
Age	52 ± 16.89	63 ± 13.59	69 ± 11.5	0.0013 ^a^
Females	45%	46%	30%	0.36 ^b^

Unless otherwise stated, the data are the means ± SD or proportions (%) or median ($\tilde{x}$) (range); ^a^: ANOVA analysis; ^b^: Fisher’s Exact test

**Table 2 ijms-23-08609-t002:** Top differentially expressed genes for each entity.

NSOI	IgG4-ROD	MALT Lymphoma
*AQP1* *, *ANG* *, *NR4A3* *, *SDC4* *, *NME4* *, *ID1*, *ITGA7* *, *PDCL3* *, *ERBB2* *, *CALD1* *, *BMPR1A* *, *TNXB* * *and RPS27A* *	*ADAM9*, *CLEC2B* *, *RNH1* *, *TLR4* *, *RBX1* *, *CLIC4. ANXA2P2* *, *RHOA* *, *NRP1*, *FCGR3A/B*, *TNFRSF1A* *, *GPX1* *, *TCEB1* *, *PSMD7*, *HSP90B1*, *TNFSF12*, *GIMAP4*, *PTGIS*, *PNPLA6*, *TGFB1*, *FCGR2A* * *and ADAM17* *	*CLIC4* *, *JAG1* *, *RAC1* *, *TNS1* *, *APP* *, *HSPB1* *, *MEG3* *, *NRP1* *, *RTN4* *, *LGALS3* *, *DST*, *CD59* *, *CFH*, *AQP1*, *ADAM15*, *AEBP1*, *ALDOA*, *ANPEP*, *ANXA2P2* * *and BMPR1A*

* Genes used in the custom-designed CodeSet for validation purposes.

## Data Availability

The data presented in this study are available on request from the corresponding author. The data are not publicly available due to local data regulations.

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
