# Peer review of "Evaluation of Orbital Lymphoproliferative and Inflammatory Disorders by Gene Expression Analysis"

_ijms, 2022, doi:10.3390/ijms23158609_

Round 1

Reviewer 1 Report

In the manuscript "Evaluation of Orbital lymphoproliferative disorders and nonspecific inflammation by Gene Expression Analysis" by Karim Al-Ghazzawi et al., the authors aim to use gene expression profiling to accurately differentiate orbital MALT lymphoma, IgG4-related disease and non-specific orbital inflammation (NSOI). The theme is interesting and extremely relevant for the field. The paper is well-written and organized. However, there is a major shortcoming towards the publication of the paper: it is based on a small amount of cases per group (n=3). It is extremely challenging to withdraw solid conclusions only from a limited amount of cases. Besides, the NSOI group is a broad one, which can include inflammatory processes from diverse etiologies. In line with this, there is a general lack of thorough information regarding the histopathological features of the cases included in the study.

Therefore, the present paper contains promising data, however, it shall only be suitable for publication if the amount of cases analyzed per group is expanded considerably.

Author Response

Response: We thank the reviewer for his careful review of our manuscript and for raising this important point to discuss. Indeed, the sample size is the main limitation, undebatable weakening the exploratory power/expressiveness of the present study. Therefore, we intensified our efforts to complete the validation of our screening results: This validation cohort comprising of 48 independent and separate samples (11 IgG4-ROD, 13 MALT-Lymphoma and 24 NSOI) has been implemented in this study. The initial results of the small sample size could be validated and show the usefulness of the potential biomarker genes.

Reviewer 2 Report

-Need to be consistent with decimals vs. commas.  (ie: line 198 uses a comma (29,3%) and line 202 uses a decimal (3.8%))

-Figure 3 is very difficult to read.  Font is extremely small and figure 4 could use a little cleanup.  Some labels are out of place.

This paper tests a small cohort of patient tissue samples for RNA to differentiate between NSOIs, IgG4- ROD and orbital MALT lymphoma by using a "conditional inference tree" algorithm.

Author Response

Comment 1.:

Need to be consistent with decimals vs. commas. (ie: line 198 uses a comma (29,3%) and line 202 uses a decimal (3.8%))

Response:  We thank the reviewer for this helpful observation and corrected the decimals

Comment 2.:

Figure 3 is very difficult to read. Font is extremely small and figure 4 could use a little cleanup.  Some labels are out of place.

Response:  We thank the reviewer for drawing our attention to the figure resolution, figures with higher resolution and therefore easier to read font were generated and implemented.

Reviewer 3 Report

I rate this as an excellent manuscript and clarifies the differences between NSOI, IgG4-ROD, and MALT lymphoma. The approach using using gene expression provides a clear indication of addressing the differences and similarities among these three disorders. I am somewhat surprised that this has not done been done before with such detail. The figures and tables provide excellent analyses of these disorders.

One thing that was a little distracting, however, is listing the genes in a text format. I would like to suggest that the genes be listed in a  form of a table, which would be easier to comprehend and compare among the three disorders. 

Author Response

We thank the reviewer for the praise and helpful suggestion. We removed the gene listing from the text and added Table 2 instead.

Round 2

Reviewer 1 Report

In the revised version of the original manuscript "Evaluation of Orbital lymphoproliferative disorders and nonspecific inflammation by Gene Expression Analysis" by Karim Al-Ghazzawi et al., the authors aim to use gene expression profiling to accurately differentiate orbital MALT lymphoma, IgG4-related disease and non-specific orbital inflammation (NSOI). The theme is interesting and extremely relevant for the field. The paper is well-written and organized. The original paper had promising data, which is now confirmed with the proposed expansion of the study. Given that the paper was considerably improved after my initial comments, I have no further objections towards its publication.

Author Response

We thank the reviewer for the praise.